# Unmet needs to admission in physical and rehabilitation inpatient department in Low and Middle Income Countries in 2023

**Rihab Moncer**[1,2,3]*, **Nedra Feni**[2,4], **Ghorbel Houssem**[1,2], **Ines Loubiri**[1,2], **Sahbi Mtaouaa**[2,4], **Sonia Jemni**[1,2], **Ahmed Ben Abdelaziz**[2,3,5]

1 Physical and Rehabilitation Department Sahloul Hospital of Sousse, Sousse, Tunisia, 2 Faculty of Medicine of Sousse, University of Sousse, Sousse, Tunisia, 3 Research Laboratory LR19SP01, Tunisia, 4 Physical and Rehabilitation Department Kairouan, Kairouan, Tunisia, 5 Information System Department, Sahloul Hospital of Sousse, Sousse, Tunisia

* rihabmoncer@hotmail.com

**Data Availability Statement:** The datasets generated and/or analyzed during the current study are available in the figshare repository at (https://

## Abstract

### Background

Determining the needs to access to rehabilitation structures is essential for developing effective improvement strategies. The objective of this study was to determine the percentage of unmet needs to admission to rehabilitation and their associated factors.

### Methods

It is a cross sectional study in the inpatient rehabilitation department Sahloul Hospital of Sousse, tertiary care center including all requests to admission. Patient demographics, diagnoses, admission decisions, and post-decision outcomes were collected for each request.

### Results

Of 329 admission requests, 316 were eligible. The mean patient age was 45 years, with a male-to-female ratio of 0.84. Most requests originated from the hospital's outpatient department, neurology, and orthopedics. Among all requests, 40.5% were not admitted. Non-neurological diagnoses and patient residency were associated with non-admission. Patients with non-neurological conditions and those residing outside the city had twice the risk of non-admission. At one month, 63% of non-admitted patients experienced functional decline, and 18% were lost to follow-up.

### Conclusion

Unmet needs to admission in rehabilitation structures is high due to lack of beds. This is leading of inequity of access to such important phase of care more. This study highlighted throwing the example of physical and rehabilitation department the gap of needs and the capacity of inpatient rehabilitation facilities. Healthcare policies should prioritize increasing rehabilitation bed availability in all regions of the country.

figshare.com/s/32286a2f6ba02136c2ec). Doi:10.6084/m9.figshare.25809061 under license CCO.

**Funding:** The author(s) received no specific funding for this work.

**Competing interests:** no competing interest.

## Introduction

The need for rehabilitation is escalating worldwide. A recent study published in The Lancet has estimated that one in three people worldwide have a medical condition that require rehabilitation [1]. The need for rehabilitation is escalating globally. A recent Lancet study underscores this, estimating that one in three people requires rehabilitation services [1]. This surge is attributed to demographic shifts and evolving health profiles. The Global Burden of Disease reveals an aging population grappling with chronic conditions like diabetes, stroke, and cancer [2], while injuries and developmental disorders persist [3]. Conversely, the paucity of dedicated rehabilitation infrastructure exacerbates the issue [4–6]. Rehabilitation, defined as optimizing function and reducing disability [7], is a cornerstone of care from diagnosis to social reintegration [8]. Inpatient Rehabilitation Facilities (IRFs) are crucial for patients with complex medical conditions and functional impairments [7, 8]. Then, project care in rehabilitation begins from diagnosis to social inclusion [8]. Patients with complex medical diseases and functional limitations need an inpatient care or as called Inpatient Rehabilitation Facilities (IRF). These departments recruit patients with stroke, neurological inflammatory diseases, spinal cord injury, brain injury, disorders of consciousness, cancer, severe burns, limb amputation, cardiac or pulmonary failure, osteoarthritis and all chronic pain conditions that influence activities of daily living [7, 8].

The World Health Organization (WHO) has emphasized the imperative of expanding rehabilitation access [9, 10]. A key indicator of accessibility is the availability of rehabilitation beds. While studies from Korea, China, and Canada have highlighted disparities in bed distribution favoring urban areas [11–13], Tunisia faces a more pronounced shortage [14]. This scarcity contributes to unmet rehabilitation needs, defined as delayed or absent essential care [15, 16]. Research on factors influencing admission to other critical care areas has identified illness severity, bed availability, and extraneous reasons [16]. However, determinants of IRF admission in Tunisia remain unexplored.

This study aims to investigate the need for IRF admission at Sahloul Hospital's Physical and Rehabilitation Department in Sousse and identify factors associated with non-admission.

## Patients and methods

Tunisia's healthcare system is a public-private hybrid, with public facilities serving specific districts. The country is divided into five districts [17]. Sahloul Hospital in Sousse is a tertiary care center offering surgical and medical services. Its Physical and Rehabilitation Medicine (PRM) department, one of two university PRM departments and the sole such facility in central Tunisia, comprises a 20-bed inpatient unit and an outpatient unit. In 2022, the inpatient unit had an 89% occupancy rate and an average length of stay of 42 days.

### Study design

This cross-sectional descriptive study encompassed patients referred for PRM admission between October 1, 2022, and September 31, 2023. A senior physiatrist evaluated each patient. The medical staff reviewed all admission requests, applying standard criteria and prioritization. Patients requiring inpatient care were admitted immediately if a bed was available; otherwise, hospitalization was scheduled. Those awaiting admission received outpatient rehabilitation services on-site or at external facilities. Patients ineligible for inpatient rehabilitation were excluded. The study population comprised two groups: admitted and non-admitted patients.

## Data collection

The following data were collected for all patients referred to PRM: age, sex, residency, diagnosis, day of contact, number of contacts, admission decision, delay between the first contact to admission, reason of non-admission, consequences of non-admission on the health situation of the patient. Age group was divided into two categories: adults from 18 to 65 years old, and the rest represented old and children. Both groups had monthly survey in our outpatient department. Outcomes of non-admission were based on the Functional Independence Measure (FIM) which is a generic scale comprising seven domains: Self-care, sphincter control, mobility, locomotion, communication and social cognition [18, 19] and 17 items. Each item is rated from one to seven according to the degree of dependence. The lowest score is 17, the highest 126. FIM was measured initially and one month later. An improvement is defined as a 20-point increase in the FIM score, and a worsening as a 20-point decrease in the FIM score.

## Statistical analysis

Descriptive statistics were used for description of the sample. The categorical variables were described as frequency and percentage and the quantitative variables were described as median, standard deviation. The chi-square test was used to compare categorical variables. Student's test was used for the comparison of a categorical variable to a quantitative variable. Univariate analysis of main variables registered at referral time was carried out. The significant level was fixed at $p<0.05$. A binary logistic regression was carried out in order to determine the factors independently associated to refusal of admission. All variables with a significance level of $<20\%$ were included in this multivariate analysis at the time of the analysis. The results were presented as odds ratios with the appropriate 95% Confidence Interval. In logistic regression model, a value of $p<0.2$ was considered as independently associated to non-admission. The significant level was fixed at $p<0.05$. The analysis was performed using the statistical package of social science (SPSS 22.0) for Windows. Geographic information were done using the Mapchart.

## Ethical status

An ethical approval was obtained from the ethic committee of faculty of medicine of Sousse. Each patient was informed about the study objective and give a written consent.

## Results

The total number of calls was 325, of which 9 were non-rational requests. Among the 316 requests left, 128 were not admitted and 188 were admitted to our department (Fig 1).

Most of requests were for women with a sex-ratio male to female 0.84. The patients included in this study had a mean age of 45.09± 18.15 years old. The age range varied between 1 and 81 years old. Among all the patients who presented for admission, 214 (67.7%) were adults and 102 (32.2%) were old and children. Of the 316 patients proposed for hospitalization, only 117 (37%) were from the government of Sousse where our IRF exists and 80% were from the third district, followed by residents of the fourth District (20%) of requests. The remaining districts one, two, and five, had a percentage of patients less than 5% (Fig 2).

Our PRM outpatient unit had the most requests 103 (32.6%), followed by neurology department 45(13.2%), orthopedics 39(12.3%), and anesthesia and resuscitation 31(9.8%) of the same hospital. Among all the requests, 80.1% were from the same hospital. The rest of the departments (22 departments), each with a percentage less than 5% (Table 1). Our

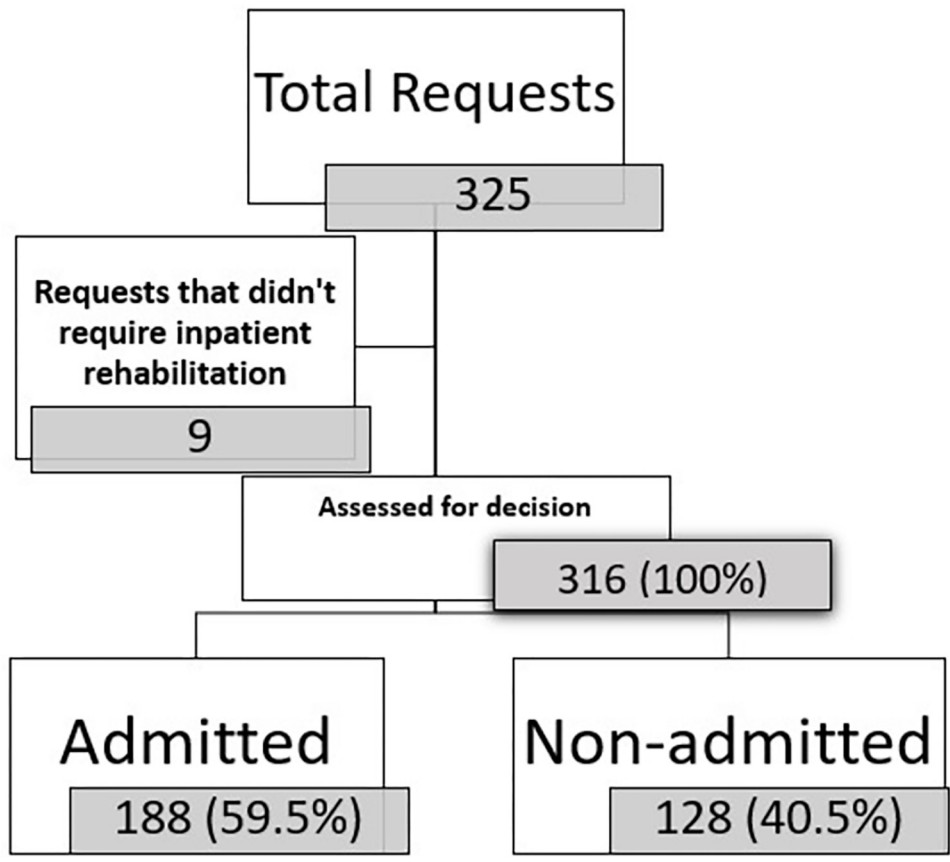

**Fig 1. Flowchart of patients included in the study unmet needs to admission in physical and rehabilitation inpatient department in Low and Middle Income Countries in 2023.**

rehabilitation department takes care of various medical pathologies. Among these, neurological and traumatic diseases were the most frequent reason for requests (**Table 1**).

The orientation of non-admitted patients was noted, of which 70 (59.8%) were referred to the outpatient clinic and 43 (36.8%) were referred to another department. When assessing functional outcome, 158 (87.8%) of admitted patients improved their FIM at baseline. Only 22 (17.2%) of the non-admitted patients improved their MIF scores, while 81 (63.0%) experienced a worsening of their MIF scores. Additionally, 24 (18.0%) non-admitted patients were lost to follow-up (**Fig 3**). In addition, 24 (18%) non admitted patients were lost to follow up. (**Fig 3**).

Most of patients with traumatic and neurological diseases were admitted (66.7%). For the other diseases, only 48.8% were admitted with a significant difference (p = 0.002).Almost 70% of admitted patients lived in Sousse while only 30% were from the other regions of the country with a significant difference (p = 0.007).Most of admitted patients were adults (63%) while almost half of non-admitted patients were old and children. There was no significant difference between admitted and non-admitted according to their age (P = 0.06).Most of males (63%) and 56% among female patients were admitted without a significant difference between sex and admission (p = 0.187).All univariate analyses are presented in **Table 2**.

This study aimed to determine the factors independently associated to non-admission of patients proposed for hospitalization in the PR department. Non neurological disease was significantly associated to non-admission (ORa = 2.069, 95% CI [1.291; 3.315], p<0.002). Also, Patients from cities out of Sousse were significantly associated to non-admission (Ora = 2.012,

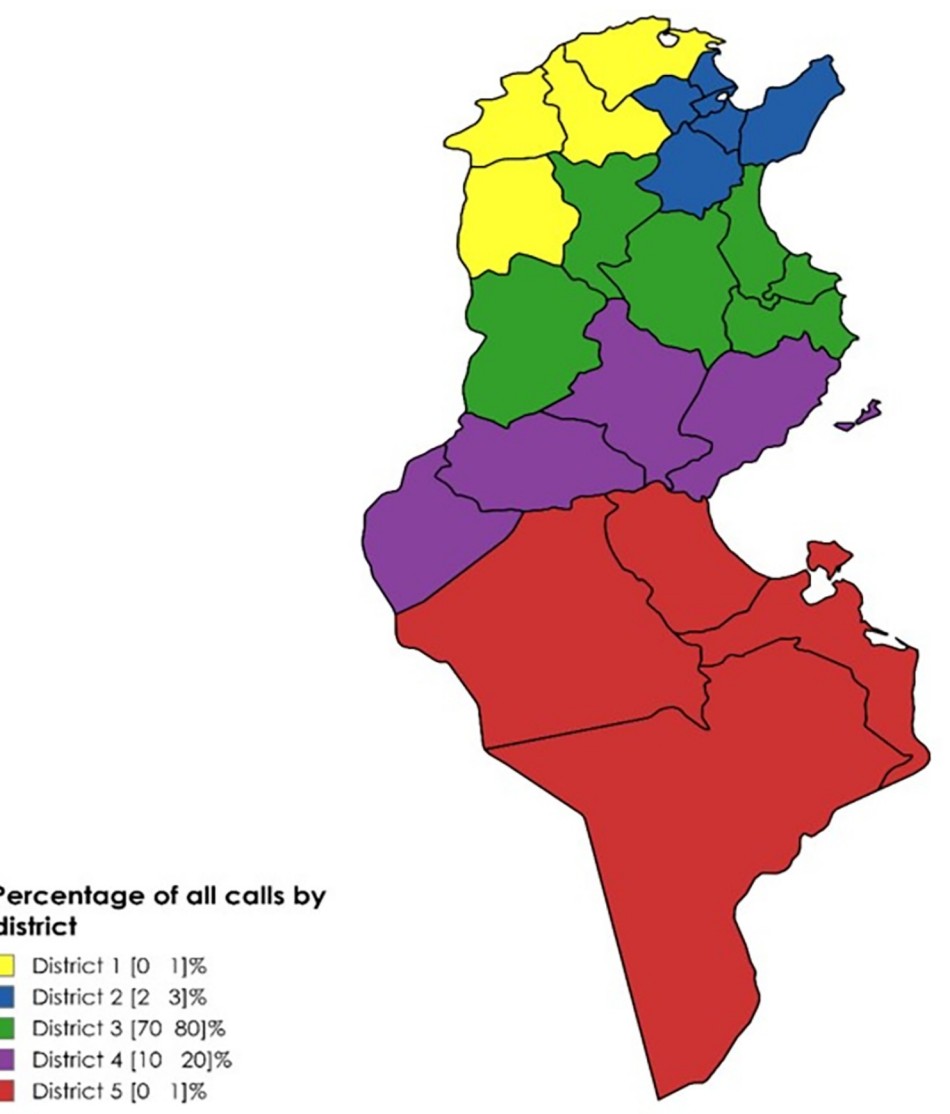

**Percentage of all calls by district**

- ☐ District 1 [0  1]%
- ☐ District 2 [2  3]%
- ☐ District 3 [70  80]%
- ☐ District 4 [10  20]%
- ☐ District 5 [0  1]%

**Fig 2. Distribution of patients addressed for admission to Physical and Rehabilitation Department Sahloul, Sousse Tunisia according to their residency.** Map was created with Mapchart.net that allowed modification and licensed under Creative Commons Attribution-ShareLike 4.0 International License (CC BY. 4.0).

95% CI [1.225; 3.307], p<0.006). Non neurological diseases and regions out of Sousse increase non admission (**Table 3**).

## Discussion

This single-center descriptive study evaluated unmet needs for admission to the Physical Medicine and Rehabilitation (PMR) department and associated factors. Nine admissions requests were ineligible for rehabilitation care. The unmet need rate was high, with nearly half of requests (40.5%) unfulfilled. Patient diagnoses and residency influenced PMR physician admission decisions. Non-neurological conditions and residency outside the IRF city doubled the non-admission risk (OR = 2). Many other medical specialties may be unfamiliar with rehabilitation hospitalization goals, explaining the requests that did not require inpatient rehabilitation.

**Table 1. Socio-demographic and clinical characteristics of population that requested inpatient rehabilitation.**

| Sex | | | Age | | | Disease | | | Department | | |
|---|---|---|---|---|---|---|---|---|---|---|---|
| N | | % | N | | % | N | | % | N | | % |
| Men | 145 | 45.7 | Adults | 214 | 67.8 | Spinal cord injury | 75 | 23.5 | PMR outpatient Unit | 103 | 32.6 |
| Women | 171 | 53.9 | Old/ Children | 102 | 32.2 | | | | Neurology department | 45 | 13.2 |
| | | | | | | Hemiplegia | 32 | 10.1 | Orthopedics department | 39 | 12.3 |
| | | | | | | Traumatic Brain injury | 27 | 8.5 | Intensive care unit | 31 | 9.8 |
| | | | | | | Low back pain and sciatica | 25 | 7.9 | Plastic surgery | 9 | 2.8 |
| | | | | | | Multiple sclerosis | 18 | 5.7 | Other PMR departments | 24 | 7.5 |
| | | | | | | Amputation | 13 | 4.1 | Other hospitals | 104 | 32.9 |
| | | | | | | Myofascial pain syndrome | 12 | 3.8 | | | |
| | | | | | | Lymphedema | 10 | 3.2 | | | |
| | | | | | | Others | 102 | 32 | | | |

N: number

PRM: Physical and Rehabilitation Medicine

Unmet needs for admission were substantial, with nearly half of inpatient unit admission requests unfulfilled. Doubling the number of beds would be necessary to accommodate all requests. This reflects a nationwide shortage of rehabilitation beds. A previous study reported an alarmingly low rehabilitation bed ratio in our country compared to developed nations, with regional disparities in access to skilled IRFs [14]. This has led to bed overbooking from various regions and increased unmet needs. Tunisia has only three public IRFs, including our department. Inequity in IRF access is a global issue, even in developed countries [11, 12]. A Korean study similarly reported disparities between Seoul and other regions [11]. Developed countries have extensively studied rehabilitation needs, with most studies demonstrating insufficient IRF capacity [20–22]. In contrast, Africa and the Middle East lack such data, necessitating updated regional assessments.

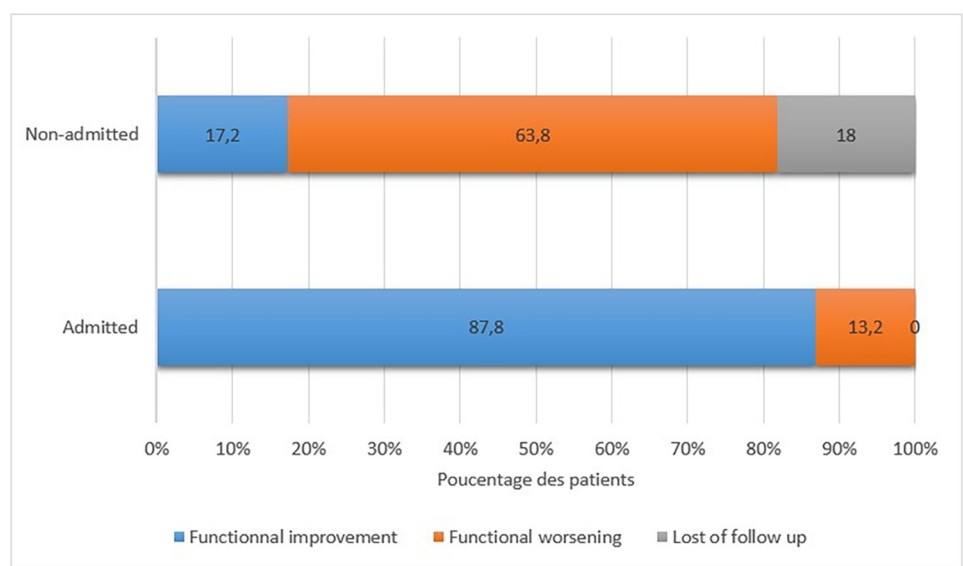

**Fig 3. Outcome of non-admitted compared to admitted patients in Physical and Rehabilitation department Sahloul Hospital of Sousse, Tunisia according to FIM scale.** FIM: Functional independence measurement.

**Table 2. Analysis of factors influencing patient admission to inpatient department in Sahloul Hospital of Sousse, Tunisia (%).**

| Factors | Group | Admitted % | Non-admitted % | P |
|---|---|---|---|---|
| **Pathology** | *Neurological* | 66.7 | 33.3 | **0.002** |
| | *Other* | 48.8 | 51.2 | |
| **Residency** | *Sousse* | 69.2 | 30.8 | **0.007** |
| | *Others* | 53.8 | 46.2 | |
| **Age** | *Old/Children* | 52.0 | 48.0 | 0.06 |
| | *Adults* | 63.1 | 36.9 | |
| **Sex** | *Male* | 63.4 | 36.6 | 0.187 |
| | *Female* | 56.1 | 43.9 | |

Research on African rehabilitation needs and unmet expectations reveals insufficient service delivery [23–25]. While epidemiological or administrative data is scarce, disability-specific surveys assessing quality of life predominate [26, 27]. Over two-thirds of disabled individuals require rehabilitation but receive no treatment, according to sub-Saharan African studies [26, 27].

The Middle East also lacks evidence, with few operational rehabilitation facilities or IRFs [6, 28, 29]. Specialized stroke rehabilitation is limited by unavailable hospital units and inconsistent care [6, 30].

To the best of the authors' knowledge, this is the first study that assessed the need for admission to IRF over a year of survey in our country, Africa and Middle East Region.

IRF hospitalization is indicated for conditions requiring intensive rehabilitation, typically three hours daily to improve patients' functional outcomes [31], to optimize functional outcomes. Target conditions include spinal cord injury, stroke, traumatic brain injury, major amputation, severe trauma, burns, systemic vasculitis with joint involvement, multiple sclerosis, Parkinson's disease, advanced osteoarthritis, knee or hip replacement, complex back/neck pain, and heart failure [8, 31–33]. Our patient screening aligned with these criteria.

Nine requests did not meet rehabilitation criteria, including nursing, palliative care, and uncomplicated low back pain. This indicates limited understanding of IRF roles among referring physicians. To address this, PMR should provide continuing professional development on conditions requiring rehabilitation. The WHO recommends training general health professionals in rehabilitation skills and promoting tele-rehabilitation to enhance access [34].

Patients with neurological and traumatic diseases residing within the same region as our IRF were prioritized for admission. This aligns with the greater participation restrictions associated with these conditions compared to musculoskeletal diseases [35, 36]. Tunisia's sector-based healthcare system further explains the preference for local patients [37, 38]. IRFs receive referrals from multiple regions, particularly the west, center, and south, suggesting a need for additional facilities in these areas. This study focused on rehabilitation bed availability as a determinant of access. While Tunisian research often addresses specific tests, sports

**Table 3. Multivariate analysis of factors influencing non-admission to inpatient rehabilitation department of Sahloul Hospital, Sousse, Tunisia.**

| Factors | aOR | IC 95% | p |
|---|---|---|---|
| Sex (Male/ Female) | 1.238 | [0.774–1.982] | 0.373 |
| Age(Old and children/Adults) | 1.561 | [0.950–2.565] | 0.079 |
| Residency (Sousse/Out of Sousse) | 2.012 | [1.225–3.307] | **0.006** |
| Disease(Non Neurological/Neurological) | 2.069 | [1.291–3.315] | **0.002** |

rehabilitation, or COVID-19 rehabilitation programs [35, 39, 40], few assess overall rehabilitation needs or services [14]. Similarly, LMICs lack rehabilitation service data, with limited capacity beyond bed shortages, including skilled personnel and assistive devices [41, 42]. The WHO reports fewer than ten skilled rehabilitation practitioners per million people in LMICs [9] and estimates that only 5–15% of those needing assistive devices receive them [43, 44]. To address these challenges, the WHO advocates for a comprehensive approach, including situation assessment, strategic planning, monitoring, evaluation, and implementation [45]. Many Eastern Mediterranean countries have developed strategic plans emphasizing collaborative efforts to expand rehabilitation services [46].

In front of these findings, increasing the number of beds in the existent IRF, creating IRF in regions with high need of admission may enhance access to rehabilitation. As shown in results, doctors choose severely disabled patients for hospitalization, and most of admitted patients are referred from acute services such as intensive care unit or neurological department. Actually, most of services of our unit are limited to post care rehabilitation. However, in developed countries, the paradigm of rehabilitation changed and hyper-specialized rehabilitation structures have been set up, with pathology-specific centers such as stroke centers, back schools, pediatric rehabilitation structures and others for the elderly [8]. It is clear that osteoarticular diseases do not require the same rehabilitative techniques as neurological disabled patients, but according to the global burden of disease, musculoskeletal conditions and low back pain are the most frequent conditions that require rehabilitation in our country and Africa [1]. As a matter of fact, it is a duty for social responsible doctors to convince health authorities and social communities to implement specific structures for those conditions. In addition, implementation of tele-rehabilitation programs to survey patients at distance may reduce the length of stay, ensure real-time monitoring of rehabilitation sessions for patients in deserted regions and detect complications earlier. Such technology has been improved during COVID-19 because of confinement and strategic approaches in many hospitals where IRF took care of post resuscitation COVID-19 patients [21]. This technology could reduce lost to follow up patients. Finally, this study is an initiation for the study of access to rehabilitation using one aspect which is the need for admission, but there are many conditions that may need rehabilitation without inpatient care. Further studies on needs of professional rehabilitation, patients' needs such as devices, specific rehabilitation such as psychmotricity, occupational therapists must be conducted in our country.

This study underscores the substantial unmet need for inpatient rehabilitation in Tunisia, exacerbated by regional disparities in bed availability. Patients with non-neurological conditions and those residing outside the IRF region face reduced admission chances. To improve rehabilitation access, expanding and strategically locating rehabilitation facilities is crucial. Evidence-based planning and implementation are essential for securing increased funding and developing effective governance frameworks.

## Acknowledgments

Authors thank all PMR Sahloul team for their engagement to enhance quality of care of admitted patients.

**Declarations:** ChatGPT 3.5 was used to correct and improve the medical writing [47].

## Author Contributions

**Conceptualization:** Rihab Moncer, Ines Loubiri, Ahmed Ben Abdelaziz.

**Data curation:** Rihab Moncer, Ghorbel Houssem, Ines Loubiri, Ahmed Ben Abdelaziz.

**Formal analysis:** Rihab Moncer, Ghorbel Houssem, Ines Loubiri, Ahmed Ben Abdelaziz.

**Investigation:** Rihab Moncer, Sahbi Mtaouaa, Ahmed Ben Abdelaziz.

**Methodology:** Rihab Moncer, Ghorbel Houssem, Sahbi Mtaouaa, Sonia Jemni, Ahmed Ben Abdelaziz.

**Project administration:** Rihab Moncer, Nedra Feni, Ghorbel Houssem, Ahmed Ben Abdelaziz.

**Resources:** Rihab Moncer, Nedra Feni, Sahbi Mtaouaa.

**Software:** Rihab Moncer, Nedra Feni, Ahmed Ben Abdelaziz.

**Supervision:** Rihab Moncer, Sahbi Mtaouaa, Sonia Jemni, Ahmed Ben Abdelaziz.

**Validation:** Rihab Moncer, Ahmed Ben Abdelaziz.

**Visualization:** Rihab Moncer, Ahmed Ben Abdelaziz.

**Writing – original draft:** Rihab Moncer.

**Writing – review & editing:** Sonia Jemni, Ahmed Ben Abdelaziz.

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
