## [Decision Letter · Decision Letter 0]

25 Jul 2024

PONE-D-24-19837Unmet needs to admission in physical and rehabilitation inpatient department in Low and Middle Income CountriesPLOS ONE

Dear Dr. Rihab,

Thank you for submitting your manuscript to PLOS ONE. After careful consideration, we feel that it has merit but does not fully meet PLOS ONE’s publication criteria as it currently stands. Therefore, we invite you to submit a revised version of the manuscript that addresses the points raised during the review process.

We look forward to receiving your revised manuscript.

Kind regards,

Barry Kweh

Academic Editor

PLOS ONE

- https://pubmed.ncbi.nlm.nih.gov/38567473/#:~:text=Results%3A%20The%20findings%20revealed%20a,East%2C%20and%20Center%20east%20districts.

In your revision ensure you cite all your sources (including your own works), and quote or rephrase any duplicated text outside the methods section. Further consideration is dependent on these concerns being addressed.

Additional Editor Comments:

A broader discussion of similar studies examining the rehabilitation programs in other countries is suggested.

Reviewers' comments:

Reviewer's Responses to Questions

**Comments to the Author**

1. Is the manuscript technically sound, and do the data support the conclusions?

Reviewer #1: Partly

Reviewer #2: Partly

Reviewer #3: Partly

2. Has the statistical analysis been performed appropriately and rigorously? 

Reviewer #1: I Don't Know

Reviewer #2: I Don't Know

Reviewer #3: I Don't Know

3. Have the authors made all data underlying the findings in their manuscript fully available?

Reviewer #1: Yes

Reviewer #2: Yes

Reviewer #3: Yes

4. Is the manuscript presented in an intelligible fashion and written in standard English?

Reviewer #1: Yes

Reviewer #2: Yes

Reviewer #3: No

5. Review Comments to the Author

Reviewer #1: The study has certain clinical significance. However, the following concerns should be addressed:

1.Is the Functional Independence Measure (FIM) suitable for all diseases?

2.The format of graphs and tables in the document needs to be optimized.

3.What are the conditions for patients who need to be hospitalized for rehabilitation? Can you tell me what the hospitalization procedure is?

4.How is your country's support for rehabilitation hospitals? Is there any corresponding policy support?

Thank you！

Reviewer #2: The manuscript would benefit from a thorough revision to improve the quality of English. Additionally, while the paper is compelling, it underscores the challenge in low and middle-income countries of admitting individuals for rehabilitation without acute illness. I suggest these countries focus on establishing dedicated rehabilitation facilities to enhance overall quality of life.

Reviewer #3: - This manuscript describes a study evaluating unmet needs of patients requesting inpatient physical and rehabilitation in a single center in Tunisia.

General:

- Language revision is required.

- Authors should read and follow author guidelines in their manuscript before submission.

Title:

- The title should represent the study. Including middle- and low-income countries in the title gives the wrong impression that this is a multinational study.

- Running title should not include abbreviations.

Abstract:

- Abstract sections do not follow guidelines (background, methods, results, conclusion).

- Having sex be presented in a ratio form is a bit confusing. This is specially that authors did not clarify if it is male/female or female /male ratio.

- Abstract section results do not show any unmet needs for admission or reasons for them being unmet.

- Conclusion has nothing to do with the findings of the study.

Patients and methods:

- Describing some of the admission requests as non-rational seems offensive. It would be better to describe the requests in a non-judgmental way.

- PRM: this abbreviation was not explained on 1st mention in the beginning of methods section. It is not until you reach the results tables that you find it in the footnotes.

- I am not sure why authors included all age groups with age range between 1-81 years. Different age groups sure have different characteristics that make their combination a problem in analysis.

Results:

- What is meant by extreme aged? Do you mean older adults, or do you mean old-old? Please describe your results more clearly.

- According to the authors description in the introduction and methods; there is a catchment area for their center. Why were there patients admitted from other areas? Please explain.

- Please explain why requests for admission for inpatient rehabilitation were sent to departments other than your department.

- In the results section, the authors stated that those admitted had improved outcome more than those who were not. The graph in figure 3 shows the opposite. Please revise.

Discussion:

- Describing not being admitted as having unmet needs made me confused. Not every patient seeking rehabilitation needs admission. That does not mean that their needs are not met, it simply means their request was declined for specific reasons. If a patient is eligible for admission and ends up not due to lack of resources; this on the other hand should be considered as unmet needs. This mix makes the aim of the work vague and overlapping with other areas.

6. PLOS authors have the option to publish the peer review history of their article (what does this mean?). If published, this will include your full peer review and any attached files.

Reviewer #1: No

Reviewer #2: No

Reviewer #3: No

---

## [Author Response · Author response to Decision Letter 0]

29 Jul 2024

Dear Editor-in-Chief of “Plos One Journal”

Thank you for giving us the opportunity to revise and resubmit our manuscript. 

We resubmit the revised manuscript title “Unmet needs to admission in physical and rehabilitation inpatient department in Low and Middle Income Countries” (PONE-D-24-19837] - [EMID:e8b087bcdc8c9ecd). 

All necessary changes were brought to this manuscript, taking into account the comments of the reviewers and the editor in chief. 

We hope that this manuscript will be eligible for publication in your reputed journal. 

Sincerely yours,

TO THE EDITOR IN CHIEF

Dear Editor in Chief,

Thank you for your comments. Please find below your responses to your questions and suggestions: Sincerely yours.

N° REMARK RESPONSE CORRECTIVE ACTIONS

 We verified the journal requirements and style, 

double space the text

adding subheading in the methods section

the style of abstract also

we included number of line in the text

we have edited the title adding the time 

also we have edited titles of figures and tables to include population, time of the study and place

- https://pubmed.ncbi.nlm.nih.gov/38567473/#:~:text=Results%3A%20The%20findings%20revealed%20a,East%2C%20and%20Center%20east%20districts.

In your revision ensure you cite all your sources (including your own works), and quote or rephrase any duplicated text outside the methods section. Further consideration is dependent on these concerns being addressed.

 Done

We putted reference in each text used in other publication 

3. Additional Editor Comments

A broader discussion of similar studies examining the rehabilitation programs in other countries is suggested. We have discussed that all studies in the regions does’nt include rehabilitation policies and other talk about lack of structure without statiscal evidence Texte L 

Line 228

TO REVIEWER #1

Dear Reviewer,

Thank you for your comments. Please find below your responses to your questions and suggestions: Sincerely yours.

N° REMARK RESPONSE CORRECTIVE ACTIONS

1. Is the Functional Independence Measure (FIM) suitable for all diseases?

Yes it is a generic scale, as we describe in the methods section, it contains 7 domain, that include motor restriction, osteoarticular restriction, communication, bowel disorder, self appearance restriction, problem solving.

That’s why we have choose it to make homogenous evaluation 

2. The format of graphs and tables in the document needs to be optimized. Thank you for this remark Done We have edited the figures, tables we have aligned some colums

3. What are the conditions for patients who need to be hospitalized for rehabilitation? Can you tell me what the hospitalization procedure is? 1- We have explained all conditions that require rehabilitation in introduction Line 61-66; it is basicly patient with total restriction of participation such as spinal cord injury, total hemiplegia, brain injury and patient with degenerative diseases that impact and limit daily living activities such as chronic low back pain and neck pain 

2- "Patients were evaluated by a senior physiatrist. Those requiring inpatient care were admitted directly if a bed was available. Otherwise, the physiatrist coordinated hospitalization when a bed became vacant. Patients not requiring inpatient care were scheduled for outpatient appointments.

Thank you for this remark and we add more explanation in the methods section 

We have added the procedure in methods section Line 95

. A senior physiatrist evaluated each patient. The medical staff reviewed all admission requests, applying standard criteria and prioritization. Patients requiring inpatient care were admitted immediately if a bed was available; otherwise, hospitalization was scheduled. Those awaiting admission received outpatient rehabilitation services on-site or at external facilities.

4. How is your country's support for rehabilitation hospitals? Is there any corresponding policy support? Before any support PMR Doctors are appealed to update their data to better plan rehabilitation accessibility in the country, Our study may be the first report 

TO REVIEWER #2

Dear Reviewer,

Thank you for your comments. Please find below your responses to your questions and suggestions: Sincerely yours.

N° REMARK RESPONSE CORRECTIVE ACTIONS

1. The manuscript would benefit from a thorough revision to improve the quality of English. Additionally, while the paper is compelling, it underscores the challenge in low and middle-income countries of admitting individuals for rehabilitation without acute illness. I suggest these countries focus on establishing dedicated rehabilitation facilities to enhance overall quality of life. .Thank you. 

.We have considered all your remarks/suggestions.

We have revised the English with an English professor initially now we used AI because of the modifications responding to the editor comments 

In declarations section we declare using AI for enhancing the langage quality

Yes we suggested at the end of the manuscript to enhance the number of beds and create new rehabilitation facilities in the country, implement telerehabilitation to increase accessibility also Line 271

TO REVIEWER #3

Dear Reviewer,

Thank you for your comments. Please find below your responses to your questions and suggestions: Sincerely yours.

N° REMARK RESPONSE CORRECTIVE ACTIONS

1. - This manuscript describes a study evaluating unmet needs of patients requesting inpatient physical and rehabilitation in a single center in Tunisia.

General:

- Language revision is required.

- Authors should read and follow author guidelines in their manuscript before submission.

Title:

- The title should represent the study. Including middle- and low-income countries in the title gives the wrong impression that this is a multinational study.

- Running title should not include abbreviations.:

 .Thank you. 

.We have considered all your remarks/suggestions.

We have putted Low and Middle Income country as this is the first study in the region and it may reflect the situation in many LMIC. Also, as we discussed the study may be valid out of Tunisia due to lack of structure in rehabilitation.

We have removed abbreviation from running title

2. Abstract:

- Abstract sections do not follow guidelines (background, methods, results, conclusion).

- Having sex be presented in a ratio form is a bit confusing. This is specially that authors did not clarify if it is male/female or female /male ratio.

- Abstract section results do not show any unmet needs for admission or reasons for them being unmet.

- Conclusion has nothing to do with the findings of the study.

 Thank you for your remark. We have corrected the sections following the guidelines

It is F/M we have put that in abstract and main text 

Line 40/ l 136

We have edited conlusion that it fit the findings 

Line 48

3. Patients and methods 

Describing some of the admission requests as non-rational seems offensive. It would be better to describe the requests in a non-judgmental way.

 Thank you for your remark. We have change that by conditions that doesn’t require inpatient rehabilitation 

Line 100, 134 and in the fig 1

4. PRM: this abbreviation was not explained on 1st mention in the beginning of methods section. It is not until you reach the results tables that you find it in the footnotes.

 We have explained the abbreviation Line Physical and Rehabilitation Medicine (PRM) 89

5. - I am not sure why authors included all age groups with age range between 1-81 years. Different age groups sure have different characteristics that make their combination a problem in analysis.

 We have not restriction for ages and our department is polyvalent, we have included all requests. Age’s charcteristics is the reflection of the population we receive. We have explained in the discussion that we lack skilled rehabilitation facilities not like developed countries. Finally, children rehabilitation does not require the same rehabilitation professional skills neither the locals. 

6. What is meant by extreme aged? Do you mean older adults, or do you mean old-old? Please describe your results more clearly.

 When analyzing our population we noticed that most of hospitalized are adults which is explained by most of injuries and neurological conditions occur in this age, so to better describe we devided our study population adults and extreme aged which mean old and children. Since it is confusing we will change it by old and children We have changed extreme aged by old and children this text 

Line 107 and in all tables

7. According to the authors description in the introduction and methods; there is a catchment area for their center. Why were there patients admitted from other areas? Please explain.

 The health policy is sectorised to prior patient from the region but if a patient form others regions needs an hospitalized, ethics oblige doctors to hospitalize him.

There are only priority which means if patients have the same conditions and the same restriction of function and there is only one bed, patient from Sousse is prioritized for admission 

8. - Please explain why requests for admission for inpatient rehabilitation were sent to departments other than your department. When there are not available beds, we sent patients to the outpatient unit, or other private unit to not deprive them from rehabilitation care because waiting time could reach 3 months. Of course, the intensity of care and number of hours of rehabilitation won’t be the same. We added this text in line to explain more 

Line 99

Those awaiting admission received outpatient rehabilitation services on-site or at external facilities

9. Results:

- In the results section, the authors stated that those admitted had improved outcome more than those who were not. The graph in figure 3 shows the opposite. Please revise.

 We have revised the figure and the text it was inversed thank you for this remark Text is revised 

Line 

151

Figure was revised 

Line 

195

10. Discussion:

Describing not being admitted as having unmet needs made me confused. Not every patient seeking rehabilitation needs admission. That does not mean that their needs are not met, it simply means their request was declined for specific reasons. 

If a patient is eligible for admission and ends up not due to lack of resources; this on the other hand should be considered as unmet needs. This mix makes the aim of the work vague and overlapping with other areas. Dear reviewer, all conditions that are included require inpatient rehabilitation, which is intensive by the number of hours of rehabilitation 3 hours minimum, therapeutic education. So, when not admitted the need is not met. 

We didn’t include conditions that can be handled in the outpatient unit such as “Stiffness of the elbow after fracture of the bones of the forearm” as example. 

All conditions are heavy and the degree of restriction of participation is important that require inpatient care. This is explained in discussion paragraph beginning Line 243

---

## [Decision Letter · Decision Letter 1]

12 Aug 2024

Unmet needs to admission in physical and rehabilitation inpatient department in Low and Middle Income Countries in 2023

PONE-D-24-19837R1

Dear Dr. Rihab,

We’re pleased to inform you that your manuscript has been judged scientifically suitable for publication and will be formally accepted for publication once it meets all outstanding technical requirements.

Kind regards,

Barry Kweh

Academic Editor

PLOS ONE

Additional Editor Comments (optional):

A well written paper which has been improved by methodological and structural corrections and provides original data on the difficulties in providing rehabilitation in less privileged countries.

Reviewers' comments:

Reviewer's Responses to Questions

**Comments to the Author**

1. If the authors have adequately addressed your comments raised in a previous round of review and you feel that this manuscript is now acceptable for publication, you may indicate that here to bypass the “Comments to the Author” section, enter your conflict of interest statement in the “Confidential to Editor” section, and submit your "Accept" recommendation.

Reviewer #1: All comments have been addressed

Reviewer #2: (No Response)

2. Is the manuscript technically sound, and do the data support the conclusions?

Reviewer #1: Partly

Reviewer #2: Yes

3. Has the statistical analysis been performed appropriately and rigorously? 

Reviewer #1: Yes

Reviewer #2: Yes

4. Have the authors made all data underlying the findings in their manuscript fully available?

Reviewer #1: Yes

Reviewer #2: Yes

5. Is the manuscript presented in an intelligible fashion and written in standard English?

Reviewer #1: Yes

Reviewer #2: Yes

6. Review Comments to the Author

Reviewer #1: The authors did a lot of work on this study. The revised manuscript has been substantial improvement. Thank you！

Reviewer #2: I think now your paper is ready to be published. It is interesting data about rehabilitation and its relevance in health care units.

7. PLOS authors have the option to publish the peer review history of their article (what does this mean?). If published, this will include your full peer review and any attached files.

Reviewer #1: No

Reviewer #2: No

---

## [Editor Report · Acceptance letter]

18 Aug 2024

PONE-D-24-19837R1 

PLOS ONE

Dear Dr. Moncer, 

I'm pleased to inform you that your manuscript has been deemed suitable for publication in PLOS ONE. Congratulations! Your manuscript is now being handed over to our production team.

Kind regards, 

on behalf of

Dr. Barry Kweh 

Academic Editor

PLOS ONE